# EquiTabPFN: A Target-Permutation Equivariant Prior Fitted Network

**Michael Arbel**[*1]    **David Salinas**[*2,3]    **Frank Hutter**[2,3,4]
[1]INRIA    [2]University of Freiburg    [3]ELLIS Institute Tübingen    [4]PriorLabs
[*]Equal contribution

## Abstract

Recent foundational models for tabular data, such as TabPFN, excel at adapting to new tasks via in-context learning, but remain constrained to a fixed, pre-defined number of target dimensions—often necessitating costly ensembling strategies. We trace this constraint to a deeper architectural shortcoming: these models lack target equivariance, so that permuting target dimension orderings alters their predictions. This deficiency gives rise to an irreducible "equivariance gap," an error term that introduces instability in predictions. We eliminate this gap by designing a fully target-equivariant architecture—ensuring permutation invariance via equivariant encoders, decoders, and a bi-attention mechanism. Empirical evaluation on standard classification benchmarks shows that, on datasets with more classes than those seen during pre-training, our model matches or surpasses existing methods while incurring lower computational overhead.

## 1 Introduction

Tabular data, a prevalent format in many real-world applications, has historically presented unique challenges to deep learning due to its lack of inherent structure compared to image or text data [10]. Foundation models, such as TabPFN [12], have recently been introduced to tackle classification tasks in tabular domains. These models leverage *in-context learning* capabilities of transformers [3], to perform both training and prediction in a single model evaluation, without requiring any parameter updates, achieving remarkable performance.

At the core of these foundational models is a pre-training procedure in which a transformer model is trained to predict test targets from test covariates, conditioned on training covariate/target pairs all of which are randomly sampled from some well-designed generative model. While it might seem surprising, at first, how a model pre-trained on synthetic data could perform well on real unseen data, recent work, such as Nagler [26], provides a theoretical study that shades some light on this phenomenon. These models, leverage the transformer architecture to perform *attention over rows*—attending to all samples simultaneously to enable cross-sample comparison. Applying attention over rows is crucial, as it enables the model to capture higher-order similarities between samples while preserving an inherent symmetry of tabular data—namely, that row order is irrelevant under the common i.i.d. assumption in supervised learning. Conveniently, such a mechanism also allows, in theory, handling an arbitrary number of training samples—a property that enables the model to generalize across tasks with varying dataset sizes without architectural modifications.

However, models such as TabPFN [12] are inherently confined to covariate–target pairs of fixed, predefined dimension, thereby limiting their applicability to datasets that match these specifications. This limitation can be alleviated via additional data pre-processing—e.g., projecting high-dimensional covariates into a lower-dimensional subspace—or by post-processing model predictions,—e.g. employing hierarchical strategies for classification tasks with many classes [31, 5]. However, the increased computational burden often offsets some of the benefits offered by *in-context learning*.

39th Conference on Neural Information Processing Systems (NeurIPS 2025).

Recent work in [24, 13] have partially addressed these challenges, allowing the model to handle arbitrary number of covariates, albeit, requiring the target dimension to have a fixed pre-defined dimension. A key insight there, is to exploit another inherent symmetry of tabular data: the arrangement of columns/covariates's dimensions should not influence model predictions. This is achieved through the bi-attention mechanism which alternates between attention over samples/rows and *columns*, i.e. covariates dimensions, thereby making the model equivariant to feature permutations just as it is equivariant to sample permutations. Nevertheless, these models remain limited to tasks where the target size matches the predefined dimensionality. While covariates are provided for both training and test samples, only training targets are available to the model. This asymmetry between training and test samples complicates direct extensions of the above approaches to handle the targets.

In this work, we propose EquiTabPFN, a novel architecture that enforces target equivariance, thus ensuring more robust and consistent predictions while handling targets of arbitrary dimensions. Unlike feature equivariance which can be directly obtained using a bi-attention mechanism, we achieve target equivariance by carefully combining three different mechanisms: (1) Bi-attention across covariates/target components and datapoints, (2) Prediction tokens to replace unavailable test targets, and (3) A non-parametric decoder preserving equivariance in predictions. We then establish the importance of target equivariance through theoretical and empirical analyses. In our theoretical study, we show that optimal functions for the pre-training procedure must necessarily be target-equivariant. Finally, we demonstrate, on real-world datasets, that target-equivariant models are beneficial both in terms of classification performance and inference runtime.

## 2 Related Work

**Prior-Fitted Networks.** Since the introduction of TabPFN in Hollmann et al. [12], which demonstrates how such a model can be successfully trained on synthetic data, several works have leveraged this architecture for applications such as Bayesian Optimization [23], forecasting [7], learning curve extrapolation [1], and fairness analysis [29]. Aside from Hollmann et al. [13], Müller et al. [24] that proposed a covariate equivariant version of the TabPFN models to better capture natural symmetries of tabular data, other works focused on improving scalability and speed of the model. This includes Müller et al. [22] who proposed to pre-train the model for producing the weights on an MLP by in-context learning, so that the resulting MLP performs well on a test portion of a particular dataset while achieving much lower latency. Recently, Qu et al. [27] improved the scalability w.r.t. to the sample size by introducing a two-stage architecture that first builds fixed-dimensional embeddings of rows, followed by a transformer for efficient in-context learning. All these approaches, still require a pre-defined fixed target dimension. The focus is orthogonal to ours, as we specifically analyze and enhance the target's representation of TabPFN family of architectures.

**Modifying Prior-Fitted Networks outputs.** To the best of our knowledge, no previous approach has proposed a *target-equivariant* architecture for foundational tabular models, with the exception of the concurrent work from Koshil et al. [16] which unlike us avoid to process labels with non-linear outputs to focus on interpretability versus accuracy. Several works have also proposed modifications to the output architecture of TabPFN. Müller et al. [24] introduced a modification of the output, replacing the linear projection from token embeddings to the target with Generalized Additive Models. This approach improves the interpretability of the model by constructing shape functions for each feature, which can be analyzed independently. Margeloiu et al. [19] and Ye et al. [36] explored combining non-parametric models on top of TabPFN. Both approaches require training a model at inference time unlike our method.

**Beyond pre-defined target dimensionality.** To address TabPFN's limitation to a predefined number of classes, Hollmann et al. [13] propose to split the classification problem into smaller ones on which the model can be employed, then to aggregate predictions using a strategy, such as the one based on an error-correcting output codes (ECOC) [5]. In this work, we show that such a strategy results in an increased computational cost compared to using our architecture that is natively target equivariant. Qu et al. [27] proposed to use a hierarchical classification strategy [31] which still incurs an increased computational cost. Recently, Wu and Bergman [35] propose a mechanism to handle arbitrary number of classes without target equivariance and require a different paradigm involving an adversarial training procedure.

**Equivariance beyond tabular methods.** Designing equivariant architectures [4] has long been recognized as beneficial, with the most prominent example being convolutional neural networks, which are equivariant to image translation [17]. More recently, research has focused on designing architectures with other symmetries, such as those present in spherical [9], set [37], or graph data [30]. Recent work has also explored incorporating symmetries in Large Language Models. For instance, Egressy and Stühmer [8] proposed a modification to self-attention that produces outputs equivariant to permutations of multiple-choice options, ensuring that the order of choices does not affect the result—a property known to be important in LLM applications such as LLM judges [39].

## 3 Background on Prior-Fitted Networks

Hollmann et al. [12] introduced a pre-trained model, TabPFN, that leverages the transformer architecture [33] to perform *in-context learning* on unseen tabular datasets for classification tasks without the need for any further training. Specifically, given training and test datasets of input-output pairs $(X, Y) := (x_n, y_n)_{n=1}^N$ and test samples $(X^\star, Y^\star) := (x_m^\star, y^\star)_{m=1}^M$, TabPFN returns a prediction $\hat{Y} = f_{X,Y}(X^\star)$, where $f_{X,Y}(X^\star)$ is the output of the network when provided with the training collection $(X, Y)$ and test queries $X^\star$. Here the input vectors $x_n$ and $x_m^\star$ belong to a euclidean space $\mathbb{R}^p$, while classes are represented by one-hot vectors $y_n \in \mathbb{R}^q$.

We now briefly describe the three modules of TabPFN model: an encoder, a backbone and a decoder, as it will help identify the main architectural constrains that impose a limit on the number of classes.

**Linear Encoder.** The encoder module constructs training and test tokens $(e_n)_{n=1}^N$ and $(e_m^\star)_{m=1}^M$ that are provided to the transformer backbone assuming the inputs $x$ and $y$ are vectors of fixed dimensions $p$ and $q_{max}$. Each training token $e_n$ is obtained by linearly embedding both covariate $x_n$ and target $y_n$ into a feature space of fixed dimension $d$ and then summing both embeddings, i.e. $e_n = Ux_n + Vy_n$, where $U$ and $V$ are trainable matrices of sizes $d \times p$ and $d \times q$. On the other hand, the test token consists only in embedding the test covariate $x_m^\star$, i.e. $e_m^\star = Ux_m^\star$ since the test target $y_m^\star$ is not provided to the network. While targets with smaller dimensions can be handled by a simple zero-padding procedure (see Hollmann et al. [12]), the encoder cannot easily accommodate target data with dimensions greater than $q$.

**Transformer backbone.** The backbone consists of a succession of residual multi-head self-attention layers between all tokens followed by a residual feed-forward network applied to each token. In order to avoid information leakage from test to train data, an attention mask ensures that all tokens can only attend to the training tokens. This also ensures that test tokens are processed independently from each other. The residual connections in the backbone preserve the initial feature dimension $d$, so that each token is still associated to a particular sample while gradually incorporating information from all training tokens.

**MLP decoder.** The decoder consists of a one-hidden layer MLP that takes each test output token $e_m^\star$ produced by the transformer backbone and produces a prediction vector $\hat{y}_m$ of dimension $q$. As the decoder requires a pre-defined target dimension $q$, it cannot be used post-hoc on new data with higher target dimensions.

A notable property of the TabPFN architecture is its invariance of the test predictions to the order by which training and test points are provided. This property is desirable since the order of training points is arbitrary and should not influence predictions on test samples. However, the network lacks *equivariance* w.r.t. the targets' dimensions, meaning that predictions are highly dependent on the order by which the training target dimensions are provided, an undesirable property as we show in our theoretical analysis in Section 5.

## 4 Target equivariant prior-fitted network

We introduce EquiTabPFN, a new model architecture for in-context learning on tabular data, that is permutation equivariant w.r.t. the target's components. Our architecture integrates self-attention mechanisms across data points and data components to leverage relationships between datapoints while preserving equivariance by processing individual attributes (such as the targets components). Unlike TabPFN which requires fixed target dimensions for all datasets, EquiTabPFN allows the dimensions of the target to change depending on the dataset as a consequence of its equivariant

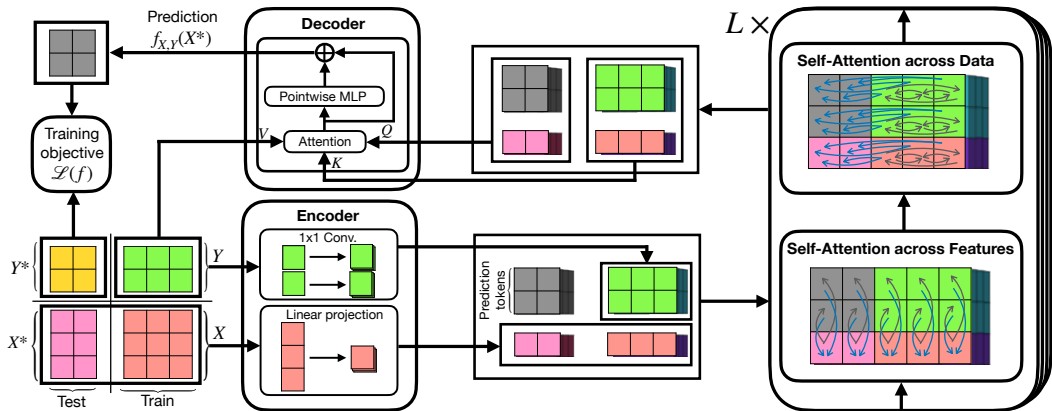

Figure 1: Overview of EquiTabPFN's architecture. Data is tokenized via an encoder, processed using self-attention, and decoded to obtain predictions. The encoder maps each covariate to a single token and embeds target components into tokens via a $1 \times 1$ convolution. Missing test tokens are replaced by prediction tokens. Self-attention alternates between (1) feature-wise attention, with target tokens attending only to covariate tokens (gray arrows) while covariate tokens attend to all tokens (blue arrows); (2) Data-wise attention, where test tokens attend only to training tokens (blue arrows), and training tokens attend to themselves (gray arrows).

architecture. EquiTabPFN consists of three major modules: a target equivariant encoder, an attention module both across data components and across data points, and a non-parametric equivariant decoder, each designed to facilitate the end-to-end learning of components interactions and datapoint relationships, see Figure 1. Below, we elaborate on each module and their interactions.

## 4.1 Target equivariant encoder

The encoder constructs training and test tokens by applying a linear projection to both covariates and targets so that target equivariance is preserved. Specifically, following [12], each training and test covariate vector $x_n$ and $x_m^\star$ is encoded into a single token of dimension $d$ by applying a linear projection matrix $U$ of size $d \times p$. However, instead of adding a linear projection of the training targets to each corresponding training token, as done in the case of TabPFN (see Section 3), we compute a token for each component $(y_n)_j$ of a training target by multiplying them with an embedding vector $V$ of dimension $d$ for all $1 \leq j \leq q$. This operation amounts to applying a $1 \times 1$ convolution along the components of each target which preserves target equivariance. Since, the validation target $Y^\star$ is not provided to the model as input, it is replaced by a trainable *prediction token* $W_{\text{pred}}$ of dimension $d$ that is repeated $M \times q$ times to form an initial guess $\tilde{Y}^0$ of the target. When considering a batch $\mathcal{B}$ of $B$ datasets $((X, Y), (X^\star, Y^\star)) \in \mathcal{B}$ of same dimensions, all these embeddings along with prediction tokens are collected to form a single tensor $E$ of shape $(B, N + M, q + 1, d)$. Here, for each batch element $((X, Y), (X^\star, Y^\star))$ of index $b$, the blocks $E_{b, :N, 1, :}$, $E_{b, N:M, 1, :}$ correspond to embeddings of $X$ and $X^\star$, $E_{b, :N, 1:q, :}$ represents the embedding of $Y$ while $E_{b, N:M, 1:q, :}$ denotes the initial guess $\tilde{Y}^0$ obtained using the prediction token. This tensor is then processed by the attention modules as described next.

## 4.2 Self-Attention Mechanisms

The core of the architecture involves two alternating self-attention modules: self-attention across components $\textbf{SelfAtt}_c$ and self-attention across datapoints $\textbf{SelfAtt}_b$ used for transforming the tokens. These alternating self-attention layers allow the model to learn both intra-samples components interactions and inter-samples relationships. Following standard design choices for transformers, we apply residual connections and layer normalization to ensure stability and robust gradient flow, i.e.:

$$E \leftarrow \text{LN}\left(E + \textbf{SelfAtt}_{c/b}(E)\right), \qquad E \leftarrow \text{LN}\left(E + \text{MLP}(E)\right),$$

where **LN** denotes the layer normalization layer [2], $\textbf{SelfAtt}_{c/b}$ denotes one of the considered self-attention mechanisms and **MLP** is a one hidden-layer network acting on each embedding independently. Below, we describe both self-attention mechanisms in more detail.

**Self-attention across components** allows interactions among components within each datapoint. It is applied independently per samples to preserve equivariance w.r.t. to the samples. In practice, this is achieved by reshaping the activation into a tensor of the shape $(B \times (N + M), q + 1, d)$ before applying attention between $q + 1$ covariate tokens of dimension $d$. We further employ a masking strategy that we found useful empirically: forcing target tokens to attend only to the covariate token, while allowing the covariate token to attend to all tokens.

**Self-Attention across datapoints** captures relationships between datapoint embeddings, allowing the model to aggregate information globally. It is applied between samples and independently per each input dimensions $p$ and $q$ to preserve equivariance. In practice, this is achieved by reshaping the activation into a tensor of the shape $(B \times (q + 1), N + M, d)$ before applying attention between $N + M$ training and validation tokens of dimension $d$. Similarly to [12], training and validation tokens only attend to training tokens.

**Remark.** EquiTabPFN and TabPFN both have linear computational complexity in the number of classes, but EquiTabPFN has a larger factor due to self-attention across components. In TabPFN, linear scaling arises from projecting classes into a fixed-dimensional space and then mapping back fixed dimensional features to classes, while the backbone remains independent of the number of classes. In contrast, EquiTabPFN's backbone scales linearly with class number, allowing it to handle arbitrary class counts.

### 4.3 Non-parametric equivariant decoder

The decoder aggregates the processed embeddings to produce prediction $\hat{Y}$. This is achieved in two steps: an attention module first computes an intermediate prediction $\tilde{Y} = (\tilde{y}_m)_{m=1}^M$ in the form of a weighted average of training targets $Y$, then a residual correction is added to produce the final prediction. More precisely, the attention module uses the embeddings of the training and validation samples as keys and queries, while the attention values are simply the training targets $Y$, i.e. $\tilde{y}_m = \sum_{n=1}^N y_n \mathrm{SoftMax}\left(\sum_{i,u} E_{b,n,i,u} E_{b,m,i,u}/\sqrt{(1+q)d}\right)$, where $b$ is the batch-index corresponding to the training target $Y$. The residual correction, in the form of a point-wise MLP, operates independently on each dimension $j$ of the attention output $\tilde{y}_m$ so that equivariance is preserved while enabling nonlinear interactions between training values.

Without the residual correction and removing the dependence of the keys and queries embeddings on the training targets $Y$ (for instance by setting the weights of the target encoder and pointwise MLP to 0), the decoder becomes a *linear non-parametric regression estimator* [32, Definition 1.7], which is a generalization of Nadaraya-Watson's estimator [25, 34]. However, linear estimators are known to be suboptimal compared to non-linear ones [6]. This motivates introducing a nonlinear dependence of the estimator to $Y$, in our setting, to increase the expressiveness of the decoder allowing it to adapt to the prediction task at hand. Experimentally, we also found a clear improvement when adding such a residual correction and making the embeddings dependent on the targets.

### 4.4 Pre-training Procedure

EquiTabPFN can be pre-trained using the same procedure as in Hollmann et al. [12], on artificial datasets sampled from a sophisticated generative model meant to capture the real-world distribution of datasets. More precisely, each artificial dataset consists of training/and test splits $(X, Y) := (x_n, y_n)_{n=1}^N$ and $(X^\star, Y^\star) := (x_m^\star, y_m^\star)_{m=1}^M$ sampled according to a conditional distribution $p(x, y|\psi)$ characterized by a *latent* parameter $\psi$. The parameter $\psi$ characterizes the dataset and is itself sampled according to a predefined prior $p(\psi)$. The pre-training procedure requires repeatedly generating artificial datasets and training the model to predict test target values $Y^\star$ given corresponding test covariates $X^\star$ as well as training covariates/target pairs $(X, Y)$ by minimizing an objective of the form:

$$\mathcal{L}(f) := \mathbb{E}\left[\ell\left(f_{X,Y}\left(X^\star\right), Y^\star\right)\right]. \tag{1}$$

Here, $(y, y') \mapsto \ell(y, y') \in \mathbb{R}$ is a point-wise loss, typically cross-entropy, and the expectation is over the collections datasets sampled according to the dataset prior. Note that each test query $x_m^\star$ is processed independently by the network $f$ so that $f_{X,Y}\left(X^\star\right) = (f_{X,Y}\left(x_m^\star\right))_{m=1}^M$. Next, we show, under natural conditions, that target equivariant functions constitute the right space of functions when searching for solutions to objectives of the form in Equation (1).

# 5 Target permutation equivariance and prior-fitted networks

We now formally analyze the impact of non-target equivariance on the training objective. We begin by precisely defining target equivariance, then show that an optimal solution must be equivariant or otherwise incurs an error quantified by the *target equivariance gap*. Finally, through empirical analysis of TabPFN training, we illustrate how the equivariance gap decreases slowly, highlighting the fundamental challenge non-equivariant architectures face in learning this key data symmetry.

## 5.1 Optimality of target equivariant networks

When presenting a new unseen dataset of covariate/target pairs $(x_n, y_n)_{n=1}^N$ to a pre-trained model, the order of the component's target is arbitrary. In other words, given target vectors of the form $y_n = ((y_n)_1, \ldots, (y_n)_q)$, these could as well be presented in a different order by applying a permutation $\sigma$ to the components of $y_n$ to obtain a permuted target $\sigma(y_n) = ((y_n)_{\sigma(1)}, \ldots, (y_n)_{\sigma(q)})$. The transformed dataset $(x_n, \sigma(y_n))_{n=1}^N$ is still essentially the same as the original dataset up to the permutation as we only changed the order of the target components. For instance, when the target represents a one hot encoding vector of 2 classes: "red" or "blue", it should not matter whether we encode "red" as the first or the second class in the one-hot vector. Consequently, a pre-trained model should be able to provide consistent predictions regardless of the component's order. More formally, the model should satisfy the following equivariance property:

**Definition 5.1** (Target permutation equivariance). *A function $f$ is permutation* equivariant *in the targets' components iff for any training data $(X, Y)$ and test covariates $X^\star$:*

$$\forall \sigma \in \mathfrak{S}_q, \quad \sigma^{-1}\left(f_{X,\sigma(Y)}(X^*)\right) = f_{X,Y}(X^*), \tag{2}$$

*where $\mathfrak{S}_q$ denotes the set of all possible permutations of the components of a vector of $q$ elements.*

It is clear from Theorem 5.1 that EquiTabPFN is target equivariant. Even when a model is not target equivariant by construction, it is natural to expect it to learn to be target equivariant, when trained via the objective in Equation (1) over a large class of randomly sampled datasets. To formalize this, we define the *target equivariance gap*, which quantifies the deviation of a function $f$ from its symmetrized counterpart $f^{\text{equi}}$.

**Definition 5.2** (Target-equivariance gap). *The target-equivariance gap $\mathcal{E}^{equi}(f)$ of a function $f$ w.r.t. to $\mathcal{L}$ is the difference between the objective values at $f$ and its* symmetrized *version $f^{equi}$:*

$$\mathcal{E}^{equi}(f) := \mathcal{L}(f) - \mathcal{L}(f^{equi}), \tag{3}$$

*where $f^{equi}$ is obtained by applying the following averaging operation w.r.t. to the uniform distribution $\mathbb{E}_\sigma$ over all permutations $\sigma$ of the target:*

$$f_{X,Y}^{equi}(X^\star) = \mathbb{E}_\sigma\left[\sigma^{-1}\left(f_{X,\sigma(Y)}(X^\star)\right)\right]. \tag{4}$$

By construction, $f^{\text{equi}}$ is permutation equivariant w.r.t. the target components. Moreover it can be easily shown that a function $f$ is itself equivariant iff $f^{\text{equi}} = f$, so that the gap vanishes. In general, the equivariance gap can take negative values. However, we establish later that this equivariance gap must be non-negative under the following assumptions on the pointwise loss $\ell$ and the marginal distribution $p$ of the data:

(A) **Invariance and convexity of the pointwise loss.** The pointwise loss $\ell$ is strictly convex in its first argument and is invariant to permutations of the components of its arguments, i.e. for any permutation $\sigma$, it holds that: $\ell(\sigma(y), \sigma(y')) = \ell(y, y')$.

(B) **Invariance of the data distribution.** The marginal distribution of the data is invariant to permutations applied to $Y$ and $Y^\star$, i.e.: $p(X, \sigma(Y), X^\star, \sigma(Y^\star)) = p(X, Y, X^\star, Y^\star)$.

Assumption (A) is satisfied for most commonly used losses such as the cross-entropy loss or the quadratic loss. Assumption (B) holds as soon as data can be presented without a preferred ordering, as most i.i.d. tabular datasets. The next proposition, proved in Section A, decomposes the objective into a non-negative equivariance gap and an optimality error.

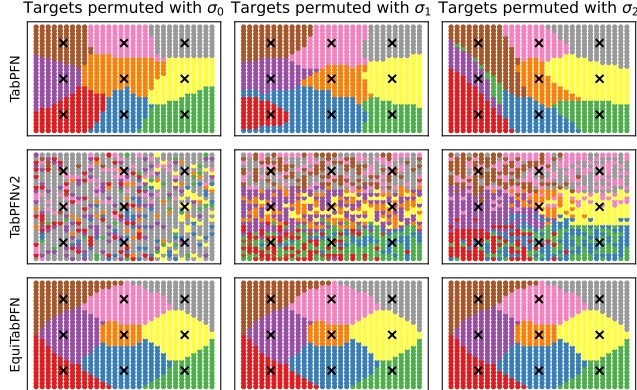
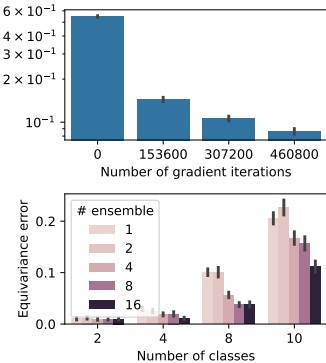

Figure 2: Prediction comparison of TabPFN, TabPFN-v2, and our model on the same datasets with three different class orderings (one per column). Models predict on a dense grid using 9 distinct training points, marked with dark crosses, each having a distinct class.

Figure 3: Equivariance error for TabPFN observed while training (top) and at inference with different number of classes and ensembles (bottom).

**Proposition 5.3.** *Under Assumptions (**A**) and (**B**), the equivariance gap $\mathcal{E}^{equi}(f)$ is always non-negative and only equal to $0$ when $f$ is equivariant to permutations, so that for any $f$:*

$$\mathcal{L}(f) = \mathcal{L}(f^{equi}) + \mathcal{E}^{equi}(f) \geq \mathcal{L}(f^{equi}).$$

*Moreover, if $f^{\star}$ is a minimizer of $\mathcal{L}$ over all measurable functions, then $f^{\star}$ must be target equivariant.*

*Proof sketch.* The key step is to express the objective as an expectation over permutations using Assumptions (**A**) and (**B**) on the data and loss: $\mathcal{L}(f) = \mathbb{E}_p\mathbb{E}_\sigma\left[\ell\left(\sigma^{-1}f_{X,\sigma(Y)}\left(X^{\star}\right), Y^{\star}\right)\right]$. The non-negativity of the equivariance gap is then established using Jensen's inequality by convexity of the loss $\ell$ in its first argument (Assumption (**A**)). Now, assume by contradiction that $f^{\star}$ is not equivariant and note that $f^{\star\,\text{equi}}$ is a measurable function by construction. It follows that $\mathcal{E}^{\text{equi}}(f^{\star}) > 0$, which directly implies that $\mathcal{L}(f^{\star}) > \mathcal{L}(f^{\star\,\text{equi}})$, thus contradicting the optimality of $f^{\star}$. $\qquad\square$

Theorem 5.3 shows that minimizing the objective $\mathcal{L}$ results in a target equivariant function. Hence, using a non-equivariant model must employ some of its expressive power solely for being equivariant, which can be wasteful, as we verify empirically in Section 5.2 below.

## 5.2 Non-equivariance of TabPFNs models

PFNs models, as introduced in Hollmann et al. [12, 13] are not permutation equivariant in the target's components. Consequently, they are not guaranteed to provide consistent predictions when the target components are permuted, thus affecting their robustness.

**Predictions instabilities.** To illustrate the implications of non-equivariance on the robustness of PFNs models, we consider a toy classification problem in 2 dimensions, where 9 training points are positioned on a 2-dimensional regular grid, each corresponding to a different class inspired by McCarter [20]. Different pretrained models are then used to predict the classes on a regular grid of $40^2$ points. Figure 2 shows the classification boundaries when using the same data but with 3 different orderings for the classes. It is clear that the ordering heavily affects the prediction even in this simple example, which strongly impacts robustness of models like TabPFN and TabPFN-v2. The predictions of TabPFN-v2 are particularly noisy due to having only 9 training data points which is not an issue in itself, unlike the extreme instability to the class ordering. Note that the axis are scaled differently for presentation.

**Target equivariance gap during training.** In Figure 3, we analyse the equivariance error while training TabPFN. Figure 3 (left) shows the equivariance error of TabPFN in terms of percentage of violation of Equation (2), e.g. how frequently the predicted classes $f_{X,\sigma(Y)}(X^{*})$ and $\sigma\left(f_{X,Y}(X^{*})\right)$ differ. We sample 512 datasets from the prior and report standard deviation. In Figure 3 (left), the equivariance error is clear and slowly decreases during training. This non-equivariance 1) induces

additional errors for the model as demonstrated in Theorem 5.3 and 2) causes the model to provide surprising predictions given that permuting the output order can change the results as seen in Figure 2.

**Mitigation via costly ensembling.** Hollmann et al. [12] introduce a mitigation strategy to non-equivariance by averaging several predictions using random permutations of the target dimensions. Averaging over all possible permutations gives an equivariant function as discussed in Equation (4). However, this requires making $\mathcal{O}(q!)$ calls to the model, where $q$ is the number of classes. This becomes quickly prohibitive, even for $q = 10$ as considered in the original study. Randomized estimators, using $N_{ens}$ random permutations can compute a prediction that converge to the averaged one at a rate of $1/\sqrt{N_{ens}}$. However, the variance of such estimator would typically present a dependence on the dimension that is, unfortunately, challenging to quantify in general. This variance is highest when the function is far from being equivariant, constituting the worst scenarios. We illustrate how fast the model becomes equivariant when using ensembles in Figure 3 (right). While ensembling more permutation helps to make the model *more* equivariant, many ensembles are required, in particular, when considering more classes. In contrast, the model we propose is fully target equivariant so that predictions remain the same regardless of the class ordering as illustrated in Figure 2 (bottom).

## 6 Experiments

### 6.1 Experimental setup

**Pretraining.** We trained our model, EquiTabPFN on artificial dataset extending the public code of Müller et al. [22] and using the same procedure from Hollmann et al. [12] employed to train TabPFNv1. Crucially, it was trained on classification tasks with less than 10 classes. The total training time took approximately 4 days on a single A100 GPU with 80GB of memory. We set our architectures hyperparameters so that they match the number of parameters of previous work, in particular the model we trained contains $\sim$25M parameters similar to the baselines we consider and we match most our hyperparameters to the same values as Hollmann et al. [12]. We refer to Section B for further training details and description of the hyperparameters used.

**Benchmarks.** For evaluation, we consider classification tasks from the TabZilla benchmark [21]. To assess the impact of unseen class counts during evaluation (i.e., exceeding 10), we consider two setups: one on 76 multi-class tasks with at most 10 classes, and another on 10 tasks with more than 10 classes. Details of the datasets/tasks used are provided in Tables 1 and 2. In all results, except those of Figure 5 (left), we report the results of EquiTabPFN without ensembling as it is not critical for the performance of our method.

**Baselines.** We consider baselines from the TabZilla benchmark. In addition, we compare with *TabPFNv2*, the state-of-the-art model with open-weights released by [13] and which was shown to out-perform standard baselines, including those appearing in the TabZilla benchmark. This model was pretrained using an improved prior compared to initial version TabPFNv1. However, the code for the prior and the training is not publicly available. Hence, to allow a fair comparaison, we also include a second version, TabPFNv2*, using the exact same architecture as TabPFNv2, but which we trained on the same publicly available prior and training code of [22] that we used for our model.

### 6.2 Main results

This section presents the main experimental findings. Additional results, including ablation of the EquiTabPFN architecture and computational cost comparisons are found in Sections C.1 and C.2.

**EquiTabPFN enables in-context-learning on datasets with unseen class counts.** Figure 4 shows accuracies relative to the KNN baseline for all models on 76 datasets with less than 10 classes (left) and 10 datasets with more than 10 classes (right). When considering datasets with more than 10 classes (right), EquiTabPFN obtains the best median relative accuracy across all datasets. It strongly outperforms TabPFNv2, which performs worse than a linear model or random forests in terms of relative median. These results show that target-equivariance allows EquiTabPFN to seamlessly generalize to larger numbers of classes even though it was trained on datasets with less than 10 classes, just like TabPFNv2. Additional evaluations in Tables 5 and 6 of Section C show that such

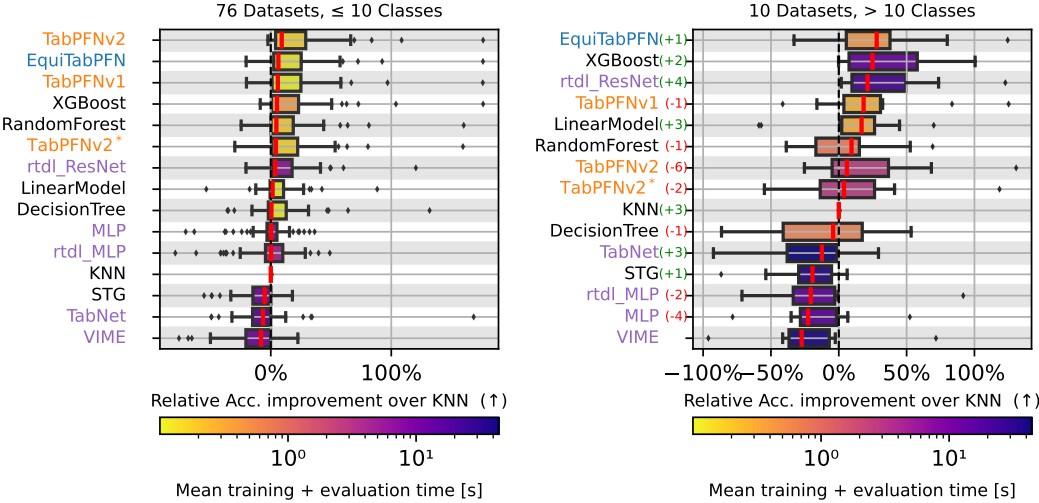

Figure 4: Relative improvement over KNN for datasets with less than 10 classes (left) and more than 10 classes (right). Red lines are the median metric over datasets after averaging each dataset over 10 splits. The runtime is displayed with color on a log scale and is reported on a V100 GPU for PFNs.

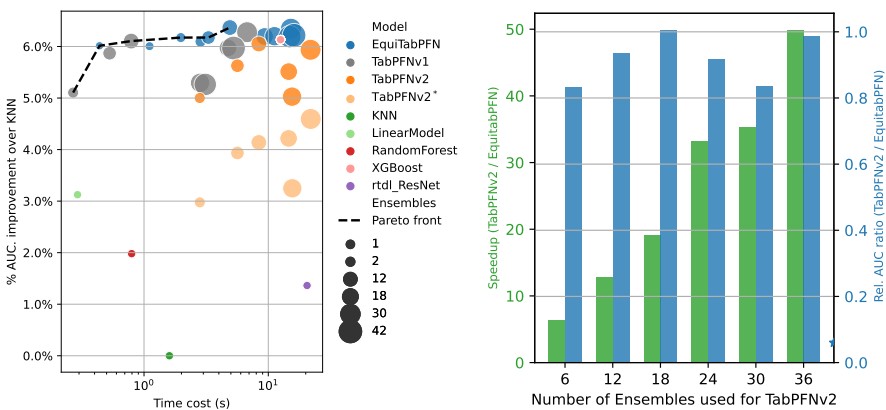

Figure 5: Left: Scatter plot of runtime vs % AUC improvement over KNN for different methods. Right: barplot of AUC ratio relatively to KNN (blue) and speedup of EquiTabPFN (green). In both figures, we increase the number of ensembles of TabPFNs variants.

improvement is consistent over various metrics: AUC, Accuracy and F1 score, for datasets with unseen class counts.

**Competitive performance on datasets with class counts seen during pre-training.** On the 76 datasets with less than 10 classes (Figure 4, left), EquiTabPFN performs comparably to the state-of-the art method, TabPFNv2, even though it did not benefit from the improved data prior used in Hollmann et al. [13] as it is not publicly available. To assess the impact of the pre-training prior on performance, we pre-trained the same network architecture as TabPFNv2 using the same publicly available protocol used for training our method and reported the results for reference (TabPFNv2*). The results show that EquiTabPFN consistently outperforms TabPFNv2* both on dataset with less than 10 classes and those with unseen class count. These results suggest that EquiTabPFN would likely benefit from the improved training procedure and prior of TabPFNv2.

**Speedup over TabPFNv2 on datasets with unseen class counts.** As discussed in Section 3, PFNs models cannot natively handle an arbitrary number of classes due to its decoder architecture. In order to apply TabPFNv2 to problems with more than 10 classes, we employ the error-correcting output codes (ECOC) strategy [5] as recommended in Hollmann et al. [13]. Such approach requires

decomposing the classification problem into several smaller classification tasks with up to 10 classes, performing predictions for each sub-task using an ensemble of TabPFNv2 models, at least one model per task, then aggregating predictions using an ECOC strategy. This incurs an extra computational cost for performing ensembling. All results reported in Figure 4 (right) are using the minimal number of ensembles to guarantee coverages of all classes. Despite this sophisticated aggregation strategy, the performance of TabPFNv2 degrades significantly, while still incurring a substantial slow-down compared to EquiTabPFN as shown by the average run-times reported by color in Figure 4 (left). Note that, for datasets with less than 10 classes (Figure 4, left) ensembling is not required anymore by TabPFNv2 and the runtime of both methods are comparable.

**EquiTabPFN achieves the best tradeoff between performance and cost on datasets with unseen class counts.** To further illustrate the improved trade-off between runtime and accuracy of our method, we show in Figure 5 (left) the improvement in AUC relatively to KNN for all methods when both TabPFNv2 and EquiTabPFN are allowed to have more ensembles. Ensembling generally improves performance of PFN models, as it allows to make more robust predictions by averaging them over transformed version of the datasets, ex: by applying permutations of the labels order as discussed in Section 5.2. While ensembling helps improve performance of TabPFNv2, the improvement is marginal compared to EquiTabPFN without ensembling and comes at a considerable computational cost, as also shown in Figure 5 (right) and on the critical difference diagrams in Figures 7 and 8 of Section C. Therefore, amongst top performing methods, EquiTabPFN achieves the best trade-off.

## 7   Conclusion

**Summary.** In this paper, we introduced EquiTabPFN, an architecture for PFNs that is equivariant to target permutations and enables in-context learning on datasets with arbitrary class counts. We established optimality of equivariant architectures for foundational tabular models and proved that non-equivariant models worsens the pre-training objective with an incompressible error term. Finally, we empirically showed the benefits of EquiTabPFN both in terms of classification performance and inference runtime on synthetic and real-world datasets. We hope this work enables future developments on prior-fitted networks that incorporate such fundamental symmetry of tabular data.

**Non-equivariance.** In some cases, the data may not be target equivariant, for instance on ordinal data. We found only a few of those cases in the benchmarks used (5 of the 86 datasets surveyed). Handling such cases would require to update the prior as it is currently equivariant to target permutation. The method could then be adapted by providing positional embedding or using column indices as input.

**Limitations.** EquiTabPFN requires a quadratic extra-cost with the number of target dimensions to perform self-attention. While many tabular problems have a relatively small number of target dimensions, this may become a problem for a very large number of dimensions and future work could consider efficient self-attention to address this issue. This may be alleviated by work focused on improving the efficiency of PFNs models [28, 38].

The code for training and evaluating our model is available at `https://github.com/MichaelArbel/EquiTabPFN/`.

## Acknowledgments

This work was supported by the ANR project BONSAI (grant ANR-23-CE23-0012-01) and was granted access to the HPC resources of IDRIS under the allocation [AD011015767] made by GENCI.

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

# A Proofs

*Proof of Theorem 5.3.* By Assumption (B) we can express $\mathcal{L}$ as an expectation of the form:

$$\mathcal{L}(f) = \mathbb{E}_p \mathbb{E}_\sigma \left[ \ell \left( f_{X,\sigma(Y)} \left( X^\star \right), \sigma(Y^\star) \right) \right]. \tag{5}$$

By Assumption (A), $\ell$ is invariant to permutations, which allows to further write:

$$\mathcal{L}(f) = \mathbb{E}_p \mathbb{E}_\sigma \left[ \ell \left( \sigma^{-1} \left( f_{X,\sigma(Y)} \left( X^\star \right) \right), Y^\star \right) \right]. \tag{6}$$

We will show that $\mathcal{E}^{\text{equi}}(f) = \mathcal{L}(f) - \mathcal{L}(f^{\text{equi}})$ is non-negative and vanishes only when $f$ is equivariant. This is a direct consequence of Jensen's inequality applied to the strictly convex function $y \mapsto \ell(y, y^\star)$ (Assumption (A)). Indeed, for any samples $(X, Y, X^\star, Y^\star)$, the following holds:

$$\ell \left( f_{X,Y}^{\text{equi}} \left( X^\star \right), Y^\star \right) : = \ell \left( \mathbb{E}_\sigma \left[ \sigma^{-1} \left( f_{X,\sigma(Y)} \left( X^\star \right) \right) \right], Y^\star \right)$$
$$\leq \mathbb{E}_\sigma \ell \left( \sigma^{-1} \left( f_{X,\sigma(Y)} \left( X^\star \right) \right), Y^\star \right),$$

where the first line follows by definition of $f^{\text{equi}}$ while the second line uses Jensen's inequality. Further taking the expectation w.r.t. $p$ shows that $\mathcal{E}^{\text{equi}}(f) \geq 0$. If $\mathcal{E}^{\text{equi}}(f) = 0$, then by the above inequality it holds that $\ell \left( f_{X,Y}^{\text{equi}} \left( X^\star \right), Y^\star \right) = \ell \left( f_{X,Y} \left( X^\star \right), Y^\star \right)$ almost surely. However, since $\ell$ is strictly convex in its first argument (Assumption (A)), the previous equality is only possible when $f^{\text{equi}} = f$ almost surely, meaning that $f$ is equivariant.

Finally, to show the final result, we note that:

$$\mathcal{E}^{\text{equi}}(f) = \mathcal{L}(f) - \mathcal{L}(f^{\text{equi}})$$
$$= \mathbb{E}_p \mathbb{E}_\sigma \left[ \| \sigma^{-1} \left( f_{X,\sigma(Y)} \left( X^\star \right) \right) - f_{X,Y}^{\text{equi}} \left( X^\star \right) + f_{X,Y}^{\text{equi}} \left( X^\star \right) - Y^\star \|^2 \right] - \mathcal{L}(f^{\text{equi}})$$
$$= \mathbb{E}_p \mathbb{E}_\sigma \left[ \| \sigma^{-1} \left( f_{X,\sigma(Y)} \left( X^\star \right) \right) - f_{X,Y}^{\text{equi}} \left( X^\star \right) \|^2 \right]$$
$$+ 2 \mathbb{E}_p \mathbb{E}_\sigma \left[ \left( \sigma^{-1} \left( f_{X,\sigma(Y)} \left( X^\star \right) \right) - f_{X,Y}^{\text{equi}} \left( X^\star \right) \right)^\top \left( f_{X,Y}^{\text{equi}} \left( X^\star \right) - Y^\star \right) \right]$$
$$= \mathbb{E}_p \mathbb{E}_\sigma \left[ \| \sigma^{-1} \left( f_{X,\sigma(Y)} \left( X^\star \right) \right) - f_{X,Y}^{\text{equi}} \left( X^\star \right) \|^2 \right]$$
$$+ 2 \underbrace{\mathbb{E}_p \left[ \left( \mathbb{E}_\sigma \left[ \sigma^{-1} \left( f_{X,\sigma(Y)} \left( X^\star \right) \right) \right] - f_{X,Y}^{\text{equi}} \left( X^\star \right) \right)^\top \left( f_{X,Y}^{\text{equi}} \left( X^\star \right) - Y^\star \right) \right]}_{=0}.$$

Here, the cross-product term equals $0$ since $\mathbb{E}_\sigma \left[ \sigma^{-1} \left( f_{X,\sigma(Y)} \left( X^\star \right) \right) \right] = f_{X,Y}^{\text{equi}} \left( X^\star \right)$ by definition of $f^{\text{equi}}$. Hence, we have shown that:

$$\mathcal{E}^{\text{equi}}(f) = \mathbb{E}_p \mathbb{E}_\sigma \left[ \| \sigma^{-1} \left( f_{X,\sigma(Y)} \left( X^\star \right) \right) - f_{X,Y}^{\text{equi}} \left( X^\star \right) \|^2 \right]$$

Finally, we use the invariance of the squared error to permutations, the equivariance of $f^{\text{equi}}$ to permutations, and the invariance of $p$ to permutations to get:

$$\mathcal{E}^{\text{equi}}(f) = \mathbb{E}_p \mathbb{E}_\sigma \left[ \| f_{X,\sigma(Y)} \left( X^\star \right) - \sigma \left( f_{X,Y}^{\text{equi}} \left( X^\star \right) \right) \|^2 \right]$$
$$= \mathbb{E}_p \mathbb{E}_\sigma \left[ \| f_{X,\sigma(Y)} \left( X^\star \right) - f_{X,\sigma(Y)}^{\text{equi}} \left( X^\star \right) \|^2 \right]$$
$$= \mathbb{E}_p \left[ \| f_{X,Y} \left( X^\star \right) - f_{X,Y}^{\text{equi}} \left( X^\star \right) \|^2 \right].$$

$\square$

# B Additional Experimental details

**Training procedure.** We use a similar training protocol as in Hollmann et al. [12], in which the model is trained on classification datasets generated according to their proposed artificial dataset prior.

In this protocol, each dataset has a fixed size of 1024 and is split into training and test uniformly at random. The maximum number of classes is fixed to 10, while the maximum dimension of the covariate vector is fixed to 100. Following Müller et al. [22], we represent the target $y$ as a one-hot encoding vector whose dimension is the number of classes in the dataset. Moreover, we employ the exact same strategy for handling missing values in the covariates. Training is performed using 153600 batches of 72 synthetically generated datasets each, which means the model was exposed to ~11M artificial datasets during pre-training, a similar order of magnitude of datasets used for pre-training TabPFN by Hollmann et al. [12]. The total training time of the network lasts approximately 4 days on a single A100 GPU with 80GB of GPU memory. The resulting network is then used for all our evaluations without altering its parameters.

We used the Adam optimizer [15] with initial learning rate of 0.0001 and linear-warmup scheduler for the first 10 epochs followed by cosine annealing [18] as in Hollmann et al. [12].

**Architecture details.** We use an EquiTabPFN network with 12 self-attention layers alternating between both type of attention introduced in Section 4: 6 blocks **SelfAtt**$_c$ and 6 blocks **SelfAtt**$_b$. Each self-attention layer consists of a multi-head attention blocks with 4 heads, embeddings of dimension 512, and hidden layers of dimension 1024. This choice ensures a fair comparison with the models used in Hollmann et al. [12], Müller et al. [22], since the number of parameters ($25.17M$) are of the same order when counting them as proposed in Kaplan et al. [14].

**Datasets.** For the datasets with less than 10 classes, we collect the ones with less than 3000 samples and 100 features and retain the datasets that contain the same number of classes accross all folds and splits. For the datasets with more than 10 classes, we filter in addition the ones able to run inference on an 80GB A100 GPU. The datasets obtained are given in Table 1 and Table 5.

| taskId | name | Classes | Features | Samples | taskId | name | Classes | Features | Samples |
|---|---|---|---|---|---|---|---|---|---|
| 3 | kr-vs-kp | 2 | 36 | 2556 | 3739 | analcatdata-chlamydia | 2 | 3 | 80 |
| 4 | labor | 2 | 16 | 45 | 3748 | transplant | 2 | 3 | 104 |
| 9 | autos | 6 | 25 | 163 | 3779 | fri-c3-100-5 | 2 | 5 | 80 |
| 11 | balance-scale | 3 | 4 | 499 | 3797 | socmob | 2 | 5 | 924 |
| 14 | mfeat-fourier | 10 | 76 | 1600 | 3902 | pc4 | 2 | 37 | 1166 |
| 15 | breast-w | 2 | 9 | 559 | 3903 | pc3 | 2 | 37 | 1249 |
| 16 | mfeat-karhunen | 10 | 64 | 1600 | 3913 | kc2 | 2 | 21 | 416 |
| 18 | mfeat-morphological | 10 | 6 | 1600 | 3917 | kc1 | 2 | 21 | 1687 |
| 22 | mfeat-zernike | 10 | 47 | 1600 | 3918 | pc1 | 2 | 21 | 887 |
| 23 | cmc | 3 | 9 | 1177 | 9946 | wdbc | 2 | 30 | 455 |
| 25 | colic | 2 | 26 | 294 | 9957 | qsar-biodeg | 2 | 41 | 843 |
| 27 | colic | 2 | 22 | 294 | 9971 | ilpd | 2 | 10 | 465 |
| 29 | credit-approval | 2 | 15 | 552 | 9978 | ozone-level-8hr | 2 | 72 | 2026 |
| 31 | credit-g | 2 | 20 | 800 | 9979 | cardiotocography | 10 | 35 | 1700 |
| 35 | dermatology | 6 | 34 | 292 | 9984 | fertility | 2 | 9 | 80 |
| 37 | diabetes | 2 | 8 | 614 | 10089 | acute-inflammations | 2 | 6 | 96 |
| 39 | sonar | 2 | 60 | 166 | 10093 | banknote-authentication | 2 | 4 | 1096 |
| 40 | glass | 6 | 9 | 170 | 10101 | blood-transfusion-service-center | 2 | 4 | 598 |
| 45 | splice | 3 | 60 | 2552 | 14954 | cylinder-bands | 2 | 37 | 432 |
| 47 | tae | 3 | 5 | 120 | 14967 | cjs | 6 | 33 | 2236 |
| 48 | heart-c | 2 | 13 | 241 | 125920 | dresses-sales | 2 | 12 | 400 |
| 49 | tic-tac-toe | 2 | 9 | 766 | 125921 | LED-display-domain-7digit | 10 | 7 | 400 |
| 50 | heart-h | 2 | 13 | 234 | 145793 | yeast | 4 | 8 | 1015 |
| 53 | vehicle | 4 | 18 | 676 | 145799 | breast-cancer | 2 | 9 | 228 |
| 54 | hepatitis | 2 | 19 | 123 | 145836 | blood-transfusion-service-center | 2 | 4 | 598 |
| 59 | iris | 3 | 4 | 120 | 145847 | hill-valley | 2 | 100 | 968 |
| 2079 | eucalyptus | 5 | 19 | 588 | 145984 | ionosphere | 2 | 34 | 280 |
| 2867 | anneal | 5 | 38 | 718 | 146024 | lung-cancer | 3 | 56 | 24 |
| 3512 | synthetic-control | 6 | 60 | 480 | 146063 | hayes-roth | 3 | 4 | 128 |
| 3540 | analcatdata-boxing1 | 2 | 3 | 96 | 146065 | monks-problems-2 | 2 | 6 | 480 |
| 3543 | irish | 2 | 5 | 400 | 146192 | car-evaluation | 4 | 21 | 1382 |
| 3549 | analcatdata-authorship | 4 | 70 | 672 | 146210 | postoperative-patient-data | 2 | 8 | 70 |
| 3560 | analcatdata-dmft | 6 | 4 | 637 | 146800 | MiceProtein | 8 | 77 | 864 |
| 3561 | profb | 2 | 9 | 536 | 146817 | steel-plates-fault | 7 | 27 | 1552 |
| 3602 | visualizing-environmental | 2 | 3 | 88 | 146818 | Australian | 2 | 14 | 552 |
| 3620 | fri-c0-100-5 | 2 | 5 | 80 | 146819 | climate-model-simulation-crashes | 2 | 18 | 432 |
| 3647 | rabe-266 | 2 | 2 | 96 | 146821 | car | 4 | 6 | 1382 |
| 3731 | visualizing-livestock | 2 | 2 | 104 | 146822 | segment | 7 | 16 | 1848 |

Table 1: List of the 76 datasets with less than 10 classes used for evaluating EquiTabPFN. The datasets are extracted from the TabZilla benchmark [21] and have a number of classes no greater than 10. Here *taskId* is the OpenML ID of the task, *Classes* indicates the number of classes, *Features* the number of covariates and *Samples* the number of samples in each dataset.

| taskId | name | Classes | Features | Samples |
|---:|---|---:|---:|---:|
| 5 | arrhythmia | 12 | 279 | 360 |
| 7 | audiology | 23 | 69 | 180 |
| 41 | soybean | 19 | 35 | 545 |
| 3022 | vowel | 11 | 12 | 792 |
| 3481 | isolet | 26 | 617 | 6237 |
| 3567 | collins | 15 | 21 | 400 |
| 3952 | chess | 18 | 6 | 22444 |
| 9956 | one-hundred-plants-texture | 100 | 64 | 1279 |
| 125922 | texture | 11 | 40 | 4400 |
| 146032 | primary-tumor | 20 | 17 | 271 |

Table 2: List of datasets used for evaluating EquiTabPFN on unseen number of classes. Datasets are extracted from theTabZilla benchmark [21] and have a number of classes greater than 10 (ranging from 11 to 100). Here *taskId* is the OpenML ID of the task, *Classes* indicates the number of classes, *Features* the number of covariates and *Samples* the number of samples in each dataset.

# C  Additional experiment results

## C.1  Ablation on the different components of EquiTabPFN.

Table 3 reports relative error reduction over TabPFN in two settings: (1) **TabPFN bb + Eq dec.** using the TabPFN backbone (without bi-attention) and our equivariant decoder, and (2) **Bi-attn bb + MLP dec.** using a bi-attention backbone and a standard MLP decoder from TabPFNv1. Since the encoder is a simple linear embedding, it is considered part of the backbone: fully connected for TabPFN-style models and 1x1 convolution for biattention-based ones. The combination of both—biattention and the equivariant decoder, *e.g*., the EquiTabPFN model we propose—yields the largest performance gain. We also tried using TabPFNv2 architecture backbone and modified it to make it target equivariant, then trained it using the publicly available code for the prior used for training TabPFNv1. This led to improvements compared to TabPFNv2* (the version we retrained ourselves using publicly available training prior). However, we found that using TabPFNv1's backbone yielded the best performance overall.

| | EquiTabPFN | TabPFN bb + Eq dec. | Bi-attn bb + MLP dec. |
|---|:---:|:---:|:---:|
| **% Error reduction** | +1.50% | +0.94% | -0.12% |

Table 3: Error reduction over TabPFN for different model configurations.

## C.2  Computational cost comparisons

**Run time comparison.** Table 4 shows the run time comparison between EquiTabPFN, TabPFNv1 and TabPFNv2 on both types of datasets (small or large number of classes). On a small number of classes, EquiTabPFN incurs a slowdown of 5x compared to TabPFNv1. This is expected as TabPFNv1 was optimized to handle data with less than 10 classes. On datasets with more than 10 classes, the gap narrows (only 1.3x slowdown) as TabPFNv1 requires ensembling techniques to handle the larger number of classes. A more complete picture accounts for the tradeoff between time cost and performance as in Figure 5 (left) and shows a clear advantage of EquiTabPFN in terms of efficiency.

| | EquiTabPFN | TabPFNv1 | TabPFNv2 |
|---|:---:|:---:|:---:|
| **< 10 Classes** | 0.12 | 0.02 | 0.16 |
| **> 10 Classes** | 0.40 | 0.30 | 2.80 |

Table 4: Time comparison (in seconds) across two types of datasets, depending on their class count.

**FLOPS comparaison.** While EquiTabPFN and TabPFN have similar parameter counts, EquiTabPFN uses more FLOPS. On an A100 GPU with 2,000 samples (100 features, 10 classes), EquiTabPFN

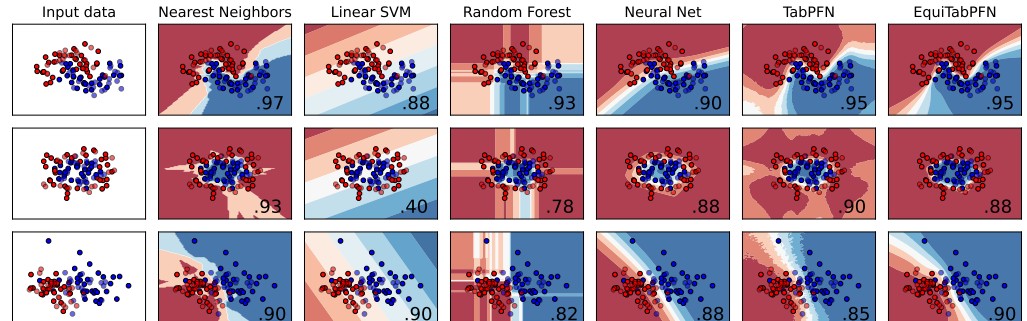

Figure 6: Binary classification decision boundary for 7 methods on 3 datasets. Even without ensembling, the boundary of EquiTabPFN is stable and smooth as opposed to TabPFN .

required 566 GFLOPS vs. 76 GFLOPS for TabPFN ($\sim$7.45$\times$ more). However, with more classes (15), the gap narrows due to the ensembling needed for TabPFN to handle more than 10 classes (EquiTabPFN: 820 GFLOPS; TabPFN: 456 GFLOPS), consistent with runtime trends. Overall, those numbers remain small for a modern GPU given that a single H100 can easily reach 400 TFLOPS on an LLM training workflow for instance.

**Memory cost.** EquiTabPFN incurs an increase compared to TabPFN which was translated in the use of smaller batch size (first dimension of the activation tensors). However, the context (in terms of the number of samples that can be processed by the model on our devices) was not affected in the experiments. This is likely due to the moderate size of the contexts used whose ranges are within the recommended limits for TabPFN.

### C.3 Binary classification decision boundary

In Figure 6, we show the decision boundary on 3 binary classification datasets for multiple baselines. To illustrate the stability of the method, we do not do ensembling for TabPFN and EquiTabPFN.

### C.4 Critical difference diagrams and performance metric tables

We show the critical diagram using Autorank implementation [11] in Figure 8 for datasets with more than 10 classes and Figure 7 for datasets with less than 10 classes. Critical diagrams show the average rank of each method (lower is better) and use a horizontal bar to show the methods statistically tied. We set the confidence level to 0.05 and use default hyper-parameters while forcing non-parametric mode to ensure stability.

We also give aggregate results for datasets with more than 10 classes in Table 5 and less than 10 classes in Table 6.

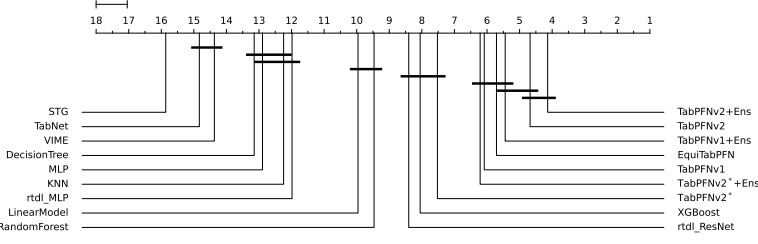

Figure 7: Critical diagram on the 76 real-world datasets with less than 10 classes from Table 1.

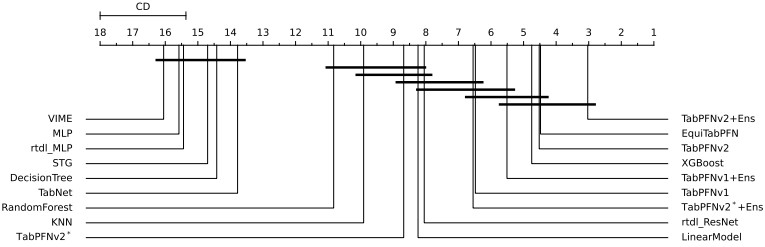

Figure 8: Critical diagram on the 10 real-world datasets with more than 10 classes from Table 2.

| model | Median relative Acc. | Mean Acc. | Mean AUC | Mean F1 | Mean time (s) |
|---|---|---|---|---|---|
| EquiTabPFN | 27.9 | 77.0 +/- 1.8 | 95.2 +/- 0.7 | 75.0 +/- 2.0 | 0.4 |
| XGBoost | 24.8 | 80.7 +/- 1.6 | 95.3 +/- 0.7 | 79.1 +/- 1.8 | 12.6 |
| rtdl(ResNet) | 21.2 | 80.9 +/- 1.7 | 91.1 +/- 1.1 | 77.9 +/- 1.9 | 20.5 |
| TabPFNv1 | 18.3 | 73.9 +/- 2.0 | 94.4 +/- 0.8 | 71.2 +/- 2.1 | 0.3 |
| LinearModel | 16.8 | 65.8 +/- 2.5 | 92.6 +/- 0.8 | 63.1 +/- 2.7 | 0.3 |
| RandomForest | 9.4 | 63.3 +/- 1.5 | 91.6 +/- 0.7 | 58.4 +/- 1.6 | 0.8 |
| TabPFNv2 | 6.0 | 74.5 +/- 2.6 | 94.4 +/- 0.9 | 73.6 +/- 2.6 | 2.8 |
| TabPFNv2* | 3.9 | 67.3 +/- 2.5 | 92.7 +/- 0.9 | 64.6 +/- 2.7 | 2.8 |
| KNN | 0.0 | 62.9 +/- 1.9 | 90.3 +/- 1.0 | 58.4 +/- 2.1 | 1.6 |
| DecisionTree | -4.2 | 51.8 +/- 1.8 | 80.7 +/- 1.0 | 47.4 +/- 1.7 | 0.7 |
| TabNet | -12.5 | 51.2 +/- 2.8 | 77.5 +/- 1.7 | 48.7 +/- 2.9 | 43.4 |
| STG | -19.3 | 46.1 +/- 1.9 | 78.7 +/- 1.1 | 39.5 +/- 1.9 | 35.2 |
| rtdl(MLP) | -20.7 | 53.9 +/- 2.5 | 74.1 +/- 1.8 | 44.7 +/- 2.9 | 13.9 |
| MLP | -22.6 | 51.0 +/- 2.1 | 73.4 +/- 1.6 | 41.6 +/- 2.3 | 15.5 |
| VIME | -27.4 | 49.2 +/- 2.6 | 70.5 +/- 1.9 | 41.3 +/- 3.0 | 39.2 |

Table 5: Aggregate accuracy, AUC, F1 and runtime for all methods on datasets with more than 10 classes. Results are ordered by the median relative accuracy improvement w.r.t KNN over the 10 datasets after averaging over the 10 different splits. Mean accuracies, AUC and F1 score are averaged over all splits and datasets. Numbers after the symbol $+/-$ refer to the standard error of the mean over all splits and datasets.

| model | Median relative Acc. | Mean Acc. | Mean AUC | Mean F1 | Mean time (s) |
|---|---|---|---|---|---|
| TabPFNv2 | 9.1 | 85.7 +/- 0.5 | 90.3 +/- 0.5 | 85.5 +/- 0.6 | 0.2 |
| EquiTabPFN | 6.3 | 83.7 +/- 0.6 | 88.9 +/- 0.5 | 83.3 +/- 0.6 | 0.1 |
| TabPFNv1 | 5.8 | 83.4 +/- 0.6 | 88.9 +/- 0.6 | 83.0 +/- 0.6 | 0.0 |
| XGBoost | 5.2 | 82.7 +/- 0.6 | 88.1 +/- 0.5 | 82.5 +/- 0.6 | 0.4 |
| RandomForest | 4.6 | 79.7 +/- 0.5 | 86.7 +/- 0.5 | 78.9 +/- 0.6 | 0.2 |
| TabPFNv2* | 4.3 | 80.8 +/- 0.6 | 87.1 +/- 0.6 | 80.1 +/- 0.6 | 0.2 |
| rtdl(ResNet) | 3.6 | 80.2 +/- 0.6 | 85.7 +/- 0.5 | 79.4 +/- 0.7 | 6.3 |
| LinearModel | 1.5 | 77.1 +/- 0.7 | 82.4 +/- 0.6 | 76.2 +/- 0.7 | 0.0 |
| DecisionTree | 0.6 | 76.2 +/- 0.6 | 79.8 +/- 0.5 | 75.1 +/- 0.6 | 0.0 |
| MLP | 0.3 | 73.2 +/- 0.7 | 76.7 +/- 0.7 | 71.0 +/- 0.8 | 6.7 |
| rtdl(MLP) | 0.3 | 73.4 +/- 0.8 | 77.3 +/- 0.7 | 71.1 +/- 0.9 | 4.8 |
| KNN | -0.0 | 73.8 +/- 0.6 | 79.5 +/- 0.6 | 73.0 +/- 0.7 | 0.0 |
| STG | -5.0 | 66.5 +/- 0.7 | 70.6 +/- 0.5 | 63.7 +/- 0.8 | 11.0 |
| TabNet | -6.4 | 68.5 +/- 0.6 | 73.5 +/- 0.5 | 67.5 +/- 0.7 | 17.2 |
| VIME | -8.1 | 63.5 +/- 0.7 | 69.7 +/- 0.7 | 60.8 +/- 0.8 | 10.5 |

Table 6: Aggregate accuracy, AUC, F1 and runtime for all methods on datasets with less than 10 classes. Results are ordered by the median relative accuracy improvement w.r.t KNN over the 10 datasets after averaging over the 10 different splits. Mean accuracies, AUC and F1 score are averaged over all splits and datasets. Numbers after the symbol $+/-$ refer to the standard error of the mean over all splits and datasets.

