# OpenReview forum: "EquiTabPFN: A Target-Permutation Equivariant Prior Fitted Network"
_NeurIPS.cc/2025/Conference — NeurIPS 2025 poster_

### Official Review · Reviewer_PBVJ · 2025-06-22

**Clarity:** 4
**Significance:** 3
**Originality:** 3
**Rating:** 5
**Confidence:** 4

**Summary:**

TabPFN is inherently limited to predicting no more than 10 classes, as Hollmann et al. (2025) observe:

> “Thus, natively our method is limited to predicting at most 10 classes.”

The authors of the current paper argue that this ceiling arises from the model’s lack of equivariance with respect to the target dimension. To overcome it, they introduce an architecture that projects each target dimension into its own token, thereby avoiding the identified weakness. On a large suite of datasets, they empirically demonstrate the resulting performance gains.

**Questions:**

1. **Figure 2.** I was surprised to see TabPFN-v2 performing worse than the original TabPFN, even though your architecture appears conceptually closer to TabPFN-v2 and achieves results closer to or better than TabPFN. Could you clarify why TabPFN-v2 lags behind TabPFN?

2. **Figure 5.** You refer to a “Pareto front,” but the concept is not explained in the text. Could you elaborate on what a Pareto front represents in this context and any consequences stemming from it?

**Ethical Concerns:**

["NO or VERY MINOR ethics concerns only"]

**Final Justification:**

The authors have answered all my questions during the rebuttal. I am confident their is significant and merits acceptance.

**Limitations:**

yes

**Quality:**

4

**Strengths And Weaknesses:**

## Strengths

1. **Significance.** The authors propose a relatively straightforward modification that performs exceptionally well in practice.
2. **Theoretical justification.** By introducing the notion of an *equivariance gap*, the authors show that methods lacking the equivariance property are theoretically bound to underperform.
3. **Extensive empirical validation.** The paper presents a comprehensive evaluation: both in simulation and on real-world dataset, highlighting key design choices, pointing out potential flaws, and reporting performance alongside computational cost for all methods considered.
4. **Clarity.** The presentation is clear and well-structured; multiple plots and tables effectively communicate the core ideas.

## Weaknesses

1. **Originality.** The final prediction layer, $\sum_{n=1}^N y_n \omega_n$ (without the residual MLP, where weights $\omega_n$ are computed via attention), together with the comparison to KNN, suggests a close relationship to nearest-neighbor methods. It would be valuable to include a comparison with ModernNCA (Ye et al., 2025), which employs a similar prediction layer, even though it is not a foundation model and must be trained per dataset.

---

Ye, Han-Jia; Huai-Hong Yin; and De-Chuan Zhan. “Modern Neighborhood Components Analysis: A Deep Tabular Baseline Two Decades Later.” *ICLR* (2025).

---

> ### Author Rebuttal · Authors · 2025-07-30
>
> ## **Comparison with ModernNCA**
> Thank you for mentioning this method. One reason why we did not include it in the comparison is that the method is not an ICL method and requires being trained on each dataset. However, we agree that it is relevant given that it proposes using a non-parametric output as in our method and we will include it in the paper.
>
> ## **Answers to questions**
>
> - **Figure 2:** We agree with you that the behavior on Figure 2 is surprising for TabPFN-v2. One reason for this behavior is that the model development considered mostly improving point prediction accuracy but has not considered the (non-)equivariant aspect to guide the model development.
>
> - **Figure 5:** Thank you for raising this point. We apologize for using this Pareto front terminology without properly introducing it, we will clarify in the paper. The Pareto front is defined by the methods that achieve the best tradeoffs of time cost vs performance (%AUC improvement) among the compared methods. These best tradeoff-achieving methods are the ones there is no other method that is strictly better both in terms of performance and time cost.
>
> Implication: The fact that EquiTabPFN  + various ensemble sizes, constitute the Pareto front means that: to achieve the highest performance given a fixed time budget, one should use EquiTabPFN + corresponding number of ensembles (closest dot on the Pareto), instead of the other models in the setup we studied.

---

> > ### Comment · Reviewer_PBVJ · 2025-08-03
> > **Response to the Rebuttal**
> >
> > Thank you for clarifying these points! I feel the paper is strong and definitely would like to see it included in the conference this year.

---

### Official Review · Reviewer_Y2PB · 2025-07-01

**Clarity:** 4
**Significance:** 3
**Originality:** 3
**Rating:** 5
**Confidence:** 5

**Summary:**

EquiTabPFN is a proposed modification to TabPFN, a transformer model that has been meta-learned to perform in-context learning based tabular classification. In particular, it fixes the lack of output class order equivariance in TabPFN, with evidence from both theoretical analysis and from tabular benchmarks that this produces meaningful improvements.

**Questions:**

Questions:

(Q1) I see that EquiTabPFN was trained with similar number of parameters as TabPFN for correctness evaluation fairness, so I am guessing that they are probably similar in computation time as well. But could you please comment more on this (and ideally include in the paper)? Did EquiTabPFN produce any slowdown in runtime? Are the computational complexities the same in terms of number of classes?

(Q2) Even though EquiTabPFN is equivariant by construction, it seems to me from Figure 1 that it's not perfectly equivariant. For example the pink and blue classes aren't identical in shape, with the pink boundary curved and the blue boundary pointy. Is this due to numerical imprecision?

(Q3) Are the x-axis and y-axis in Figure 1 equally scaled? It's not clear to me what are the sets of classes that should be identical in decision boundary shape to each other.

(Q4) Related to (W1), is there a way to reengineer non-equivariance back into the model, the same way position encodings allow one to "learn to bypass" attention's permutation-invariance?

(Q5) Analysis about which datasets in the benchmark improved or worsened from TabPFN to EquiTabPFN would be interesting. Were there any datasets that have some kind of class-order non-equivariance that TabPFN manages to exploit?

**Ethical Concerns:**

["NO or VERY MINOR ethics concerns only"]

**Final Justification:**

Reading the other reviews and the authors' rebuttal strengthened my belief that this submission warrants acceptance into NeurIPS. In particular, the authors' responses allayed my concerns about this work being incompatible with belief in The Bitter Lesson. I am not changing my original score because I already gave it a 5 (accept) with confidence 5 (full confidence). I didn't assign a 6 (strong accept) because I'm not sure about "groundbreaking impact", but I think this paper is indeed "technically flawless" with likely significant impact.

**Limitations:**

yes

**Quality:**

4

**Strengths And Weaknesses:**

Strengths

(S1) [Significance, Originality] The presented problem (lack of equivariance for permutions of classes in TabPFN) and proposed solution (a classification head that is class order equivariant) are relevant, clever, and interesting.

(S2) [Significance, Quality, Originality] The theoretical analysis showing the benefit of equivariance is very nice. Instead of being a box-checking exercise or an impress-the-reviewers exercise, the analysis proves something interesting and contributes to the narrative.

(S3) [Significance, Quality] The results on both the initial analyses and also on the benchmarks are great. It's surprising that ensembling in TabPFN fails so badly to attain (approximate) equivariance, and that the very rough decision boundaries in Figure 1 would be solved by equivariance.

(S4) [Clarity] The paper is very clearly written.

Weaknesses

(W1) [Significance] Code was not provided, but the authors explained that anonymization wasn't ready at time of submission. So we hope that this can be addressed by the rebuttal period.

(W2) [What follows is a subjective, debatable point, and is not a reason for rejection.] Efforts to introduce invariances / equivariances have generally faced the Bitter Lesson. For example, the most recent version of AlphaFold found that dropping rotational equivariance offered better performance. Why should we not suspect that EquiTabPFN won't meet the same fate? For example, one might imagine that in many cases, modelers sort their classes (with say the first one being the "default" majority class). As another example, certain classification problemsPerhaps a non-equivariant model will be able to learn and exploit such phenomena while defaulting to equivariance.

---

> ### Author Rebuttal · Authors · 2025-07-30
>
> Thank you for your positive feedback. We provide below answers to your questions, hoping they address them.
> ## **Code availability**
> We unfortunately cannot share the code at the rebuttal as submitting any link is forbidden by the current Neurips policy. We are fully committed to share all the code necessary to reproduce our experiments with the paper including the code of the method and the code to reproduce figures and tables.
> ## **Equivariance vs no equivariance**
> This is a great point that we think should be emphasized in the paper. Here are our current thoughts on the question.
> - **What does equivariance buy for TabPFN?** It allows the model to handle an arbitrary number of classes, which is one of the key strengths of the model. This property would not be true even if the original non-equivariant was trained to achieve 0 equivariance error (for a fixed maximal class number). That being said, one could retain this property (handling arbitrary class number), while breaking equivariance in a controlled way, if it is not always desirable. For example, by adding a form of positional encoding as you suggested (see also answer to Q4). Hence, equivariance, could be a general guiding principle for building models that naturally adapts to varying sizes and shapes. Once constructed, one could choose to break equivariance in a model using general recipes (ex: positional encodings).
> - **The case of AlphaFold 3:**  It is unclear from the paper whether the performance boost is due to absence of equivariance, or the use of a better modeling approach (diffusion model instead of an equivariant structure model). The authors explained that they do not use an equivariant diffusion model to simplify the architecture, but they did not claim that using an equivariant diffusion model performed worse. They motivate the use of a diffusion model by noting that the complexity of the structure module in AF2 had only a minor effect on performance and that diffusion models avoid the ‘torsion-based parametrization’ and explicit ‘violation losses on the structure’ present in AF2.
>
> ## **Q1: Computational cost**
>  - **Run time comparison:** Thank you, we will add those and discuss them in more detail based on the table below showing the average times on the datasets considered in the experiments (either more than 10 classes or less than 10 classes). On datasets with a small number of classes, EquiTabPFN incurs a slowdown of 5x compared to TabPFNv1. This is expected as TabPFNv1 was optimized to handle data with less than 10 classes. On datasets with more than 10 classes, the gap narrows (only 1.3x slowdown) as TabPFNv1 requires ensembling techniques to handle the larger number of classes. A more complete picture accounts for the tradeoff between time cost and performance in figure 5, to which we added TabPFNv1 as requested by reviewer gnTN. It shows that, given a time budget,  EquiTabPFN remains the most efficient method most of the time due to the fact that TabPFNv1 typically requires more ensembles to reach good performance likely due to its non-equivariance as illustrated in Figure 3).
>
>
> | Model        | < 10 Classes | > 10 Classes |
> |--------------|--------------|---------------|
> | EquiTabPFN   | 0.12    | 0.4           |
> | TabPFNv1     | 0.02     | 0.3           |
> | TabPFNv2     | 0.16     | 2.8           |
>
>
>
>
> - **Complexity:** The theoretical computational complexity in the number of classes is linear for both, but with a larger factor for EquiTabPFN. The linear dependence for TabPFN comes from projecting the classes into a fixed dimensional vector and then mapping back a fixed dimensional feature to the number of classes, while the dependence on the class number is constant for its backbone.  For EquiTabPFN, the backbone features scale linearly in the class number, but it enables handling arbitrary class numbers.
> ## **Q2 and Q3: On figure 2 (robustness to class ordering)**
> From Q2 and Q3, we understand that our current presentation of the figure suggests there could be some symmetries in the decision boundary between classes that the network should capture. We do not expect EquiTabPFN to exhibit such symmetries, as nothing in the model constrains it, e.g. blue vs pink clusters need not have the same shape as nothing would enforce the shape to capture the Voronoi diagram of the 9 training points). We do expect, however, that EquiTabPFN produces the same decision boundaries across the bottom three figures as a consequence of target equivariance (i.e. after permuting the order of the classes numbers). This is indeed the case, up to numerical precision. Thank you for raising this point, we will make sure to clarify this.
> Q3: Scaling in figure 2: We will clarify that both axes do not have the same scaling.
> ## **Q4: Reengineering non-equivariance back into the model**
> This is a great question! Also related to one from reviewer `gntN`.  Indeed, one could think about breaking equi-variance by providing some form of positional encoding. One would have to adapt the prior to generate non-equivariant but realistic datasets (currently only equivariant datasets are generated). One could then use column index or name, to define the encoding and consider ordinal classes. We believe this can be a promising future work direction.
> ## **Q5: Analysis per datasets**
> Most of the datasets have nominal classes which are equivariant. For the 5/86 datasets that are ordinal, we could not see a particular difference for TabPFN performance. As noted above, we believe it is expected as TabPFN is trained with an equivariant prior so it is unlikely it would be able to exploit non-equivariant information as it was not trained to handle it.

---

> > ### Comment · Reviewer_Y2PB · 2025-08-01
> > **Re: Rebuttal by Authors**
> >
> > Thanks for the thorough and helpful explanations!

---

> ### Comment · Reviewer_Y2PB · 2025-08-07
> **One more question: why is equivariance not used for language modeling**
>
> Sorry, I have an additional question. What is the intuition for why this type of equivariance is not used in LLMs? After all, with next-token-prediction, you are performing classification where the number of classes (the vocabulary size) is extremely large, and the ordering of tokens in the vocabulary / embedding table is arbitrary. Assuming that class equivariance has not been attempted for language modeling, why not? Or, if you're aware of previous works where this has been attempted, were there problems scaling this to hundreds of thousands of classes, such that it wasn't worth it? Or perhaps, is the proposed method possibly useful for language modeling?

---

> > ### Author Response · Authors · 2025-08-07
> > **answer**
> >
> > Thank you for the interesting question!
> >
> > In term of motivation, the closest paper in LLMs is this one in our opinion [1]. It addresses equivariance in the cases where sets of inputs are presented to an LLM, for instance retrieved passages for RAG or completions to compare for LLM-judges. We came aware only after the deadline but we will make sure to include it.
> >
> > Generally equivariance has been applied to text models quite a bit in the context of compositionality, see for instance [2] which proposed a way to enforce group-equivariance to ensure more are compositional when permuting certain entities.
> >
> > Regarding your more general point about the equivariance on the class order in LLMs. One challenge would be as you say to scale to hundreds of thousands of classes for the vocabulary size. We note that permuting classes means something different for a model that is trained with a fix order like a LLM (which always sees the same class for a given token) and a ICL model where the class order is sampled randomly during the procedure. Our method would not be applicable directly for LLMs given this.
> >
> > Thank you again for the interesting question, we will include a discussion of the case of equivariance for LLMs in our related work.
> >
> > [1] https://arxiv.org/pdf/2505.15433
> > [2] https://openreview.net/forum?id=SylVNerFvr

---

> > > ### Comment · Reviewer_Y2PB · 2025-08-07
> > > **Thanks for answer!**
> > >
> > > Thanks for the helpful answer. I agree that a brief summary of your answer would be helpful to include in the final version.

---

### Official Review · Reviewer_gntN · 2025-07-01

**Clarity:** 3
**Significance:** 3
**Originality:** 3
**Rating:** 4
**Confidence:** 4

**Summary:**

The authors propose a mechanism to make TabPFN-style models equivariant to the ordering of the class labels, something which is often achieved in practiced by randomizing over permutations of those.
The authors methods is to essentially tie a vector to each class dimension and perform a cross attention at the last layers to get similarity between the "validation" token refined through the forward pass and the class tokens. This also allows them to extend naturally beyond a fixed number of classes which is desirable.

**Questions:**

See above

**Ethical Concerns:**

["NO or VERY MINOR ethics concerns only"]

**Final Justification:**

The work is overall well executed but I still have reservations about the significant compute required by the method when used with the rows-as-tokens architecture as K times more tokens/compute units/memory units are now allocated to the class module.

**Limitations:**

yes

**Quality:**

3

**Strengths And Weaknesses:**

Simple architectural improvements that can improve the models are always welcome, I have a few questions and reservations.

Q1: To make sure I understand correctly, your activations vector are of shape $(B, N+M, q+1, d)$? For the instance attention you reshape to $(B*(q+1), N+M, d)$ where batch size is first, then transformer sequence length, then transformer dimension. For the self attention across labels dims/features, it would be $(B*(N+M), q+1, d)$?

The method overall makes sense, and so does the cross-attention to compute outputs.
On the evaluation side, I appreciate the ablations/baselines with variants of TaPFN v1/2 to disentangle the impact of different choices.

Q2: I see that you have tried to be faithful to TabPFN v1 procedure and hyperparameters, so would it be fair to say that using your training code but using TabPFN v1 architecture you would get very similar experimental results to TabPFN v1? I am asking because some small choices (adding a cosine in the prior bank of activations, changing the some optimization parameters...) can affect/improve the performance of those model, and as such I am not totally certain I can use TabPFN v1's performance as a reference point for equitabpfn.

Q3: I am not too sure how a 9 instances predictions problem is really all that insightful but the vizualization is nice. However as TabPFN v1 is typically averaged over class permutation, does this averaging actually lead to decent clusters?

Q4: could you comment on Figure 7 and 8 in appendix? it seems that for <=10 classes equitabpfn has a performance between tabpfn v1 and tabpfn v1 ensemble?

Q5: Apologies if I missed but have you tried your method on top of tabpfn v2? What are the results?

Q6: *Computational efficiency and memory* Finally this is my most important remark and question. If I understand correctly, compared to tabpfn v1, you now deal with a 4D tensor, and as such your activations are much larger (q+1 times) which is bound to limit the context size you can use. Does your method exhibit a reduced context size compared to v1?
Furthermore, it does also increase the computations if I understand correctly. Where is TabPFN v1 and TabPFN v1-ensemble in the time/performance graph? It seems to me that for the same memory and computational cost you can afford to have the vanilla more efficient architecture+some bagging for free.

Lastly, this is a more general open question on which I am interested to have your opinion. I think being able to extend beyond a fixed number of classes is great. I am not convinced baking equivariance in the model is necessary. In the same way that Alphafold 2 simply used data augmentation over many protein orientations instead of baking the equivariance in the model, a sufficiently large model is able to learn invariance/equivariance on its own. Even more that this, there may be some patterns in the training data that could be destroyed by equivariant/invariant transformations. In tabular data, classes are actually often ordinals in disguise (age, number of children etc...) so I tend to believe that just showing real data to a sufficiently powerful model would enable it not only to be invariant to order but also *when* to be.

## Minor points
Line 276: While indeed O(q!) would be needed for perfect invariance to ordering, in practice the prediction would concentrate with O(1/sqrt(N_ensembles)) which should converge on a given class in actually a not so high number of ensembles.

---

> ### Author Rebuttal · Authors · 2025-07-30
>
> Thank you for your encouraging and thorough feedback. To address your question on time/performance comparison,  we included TabPFN v1 in figure 5 and present the pareto front in table format (below). Below, we address your questions.
> ## **Q1 – Tensor shapes**
> Yes, the tensor shapes are modified as you say. We will add an explanation using shapes in the text for clarity.
> ## **Q2  – Reproducing TabPFNv1**
> Yes, we reused the public reimplementation from the Mothernet paper and we confirm that we could reproduce the results of TabPFNv1 using this setup which resulted in similar performance as the publicly available pre-trained model from the original author paper. We kept the optimization hyperparameters as-is and did not change nor tune them.
> ## **Q3  – Averaging TabPFNv1 in figure 2**
> Yes indeed, TabPFN-V1 converges to better clusters when increasing the number of ensembles. However, as shown in Figure 3, the equivariance error decreases only slowly and the equivariance error is still significant when using 16 ensembles for instance.
> ## **Q4  –  Critical diagrams (figures 7 & 8)**
> - **Figure 7:** Ensembling mechanically helps increase TabPFNv1 performance, but induces extra cost. Please also refer to the updated time/comparison results  (answer to Q6), which compares TabPFNv1 with different ensembling to EquiTabPFN.
> - **Figure 8:** Similarly, ensembling increases performance of TabPFNv2 at an extra cost. This is also consistent with the performance vs time cost  plot in figure 5 (left).
> Overall these results are consistent with the claim of the paper: EquiTabPFN deals more efficiently (better performance at lower time cost) with an unseen number of classes thanks to its target equivariance architecture.
>
> ## **Q5  –  Using TabPFNv2 backbone**
> We tried TabPFNv2 architecture backbone and modified it to make it target equivariant, then trained it using the publicly available code for the prior used for training TabPFNv1. This led to improvements compared TabPFNv2* (the version we retrained ourselves using publicly available training prior). However, we found that using TabPFNv1’s backbone yielded the best performance overall. We will clarify that in the text.
>
> ## **Q6  –  Computational efficiency and memory**
> - **Effect of architecture on context size:** Indeed, there is an increase in memory cost, this had an impact on the batch size (first dimension) that we had to reduce during training. However, the context (in terms of the number of samples that can be processed by the model on our devices) was not affected in the experiments. This is likely due to the moderate size of the contexts used whose ranges are within the recommended limits for TabPFN. We agree however, that scaling to larger datasets could cause memory issues. We will clarify this in the text.
> - **Time/performance comparison with TabPFNv1:** Thank you for this comment, it is an excellent point! The focus of the time/performance graph was on TabPFNv2 as the SOTA method. However, as you pointed out, TabPFNv1 is lighter and would benefit from additional ensembling. When adding TabPFNv1 and EquiTabPFN* (the version using TabPFNv2 backbone) to the time/performance graph, we observed that TabPFNv1 indeed contributes (2) points on the Pareto front (2 points), while the points contributed by EquiTabPFN to the front remain unchanged. Hence, EquiTabPFN remains efficient on a wide range of time budgets. As we cannot include the graph here, we present a table summarizing these results. We will add the graph and discussion in the revised version.
> **Time/performance tradeoff**:
> | Pareto points   | 1        | 2        | 3        | 4        | 5        | 6        | 7        |
> |---------|----------|----------|----------|----------|----------|----------|----------|
> | Old Pareto front   | LinearModel     | EquiTabPFN(1)       |___________| EquiTabPFN(4)      | EquiTabPFN(12)      | EquiTabPFN(18)        | ___________ |
> | New Pareto front | TabPFNv1(6)         | EquiTabPFN(1)       | TabPFNv1(18)       | EquiTabPFN(4)       | EquiTabPFN(12)        | EquiTabPFN(18)    | EquiTabPFN*(18)   |
> | Time| 0.269 | 0.439 | 0.789 | 1.978 | 3.230 | 4.907 | 9.979 |
> | Performance| 51.06 | 60.14 | 61.06 | 61.73 | 61.74 | 63.68 | 66.50 |
>
> The table below shows how the pareto front is affected by adding TabPFNv1 and EquiTabPFN* (the version using TabPFNv2 backbone). The first row is the order of the methods crossing the pareto front from left to right.  Four observations:
> 1. Overall, the points forming the pareto and corresponding to EquiTabPFN are unchanged.
> 2. The leftmost point on the front is no longer LinearModel and is now TabPFNv1 using an ensemble of size 6 (smallest possible to handle more than 10 classes as discussed in the paper).
> 3. There is an additional point obtained using TabPFNv1 with 18 ensembles, that is inserted between EquitabPFN (1 ensemble) and EquiTabPFN (4 ensembles).
> 4. An additional point obtained using EquiTabPFN* with 18 ensembles is added at the rightmost side.
>
> ## **General question: Equivariance vs no equivariance**
> We thank you for this great question, we will discuss this point in the paper. Our thoughts:
>
> - **When to (not) be equivariant:**  We agree that if data is not exchangeable and has additional structure, e.g. ordinal, one should not aim at enforcing equivariance (which would destroy it), and instead leverage the ordinal structure. However, the prior used in TabPFN is equivariant and because of that, the model still ends up learning approximate invariance during training as you pointed out (also shown in Figure 3).
> - **Controlling (non)-equivariance:** It is unclear to us to what extent current models (such as TabPFN) could learn a mechanism to control when to be equivariant given that they are trained with an equivariant prior. However, if equivariance is no longer desirable, one could think of a way for building such a control mechanism in EquiTabPFN to break equivariance while retaining the ability to handle any class number. As pointed out by reviewer Y2PB, a  “positional encoding” computed at inference time could be added to the tokens class to differentiate them from each other. When providing the same positional encoding to all tokens, the representation becomes equivariant, otherwise, it is not. For this, the prior would have to be adapted to include examples of non-equivariant and realistic artificial datasets which we believe is an interesting research direction.
>
> ## **Minor points: Estimation by ensembling**
> Indeed, we will clarify that point. The variance of the estimator could present a dependence on the dimension, however it is challenging for a general function. The closer the function is to equivariance the smaller such variance would be. Hence, the worst scenarios happen when the function is far from being equivariant, in which case, the variance is the highest.

---

> > ### Comment · Reviewer_gntN · 2025-08-04
> > **answer to rebuttal**
> >
> > Thank you for your thoughtful answer and the additional experiments, the experimental setting overall looks strong to me.
> > Concerning the additional memory/compute added the class dimension (4d tensors), I understand increasing aggregation is feasible for the parameters of TabPFN v1 (1024 max ctx+qy size). However, a significant drawback of these ICL TFMs is the decrease in performance on large datasets due to limited context size possible. It also seems that long-context ICL models is currently an important direction being investigated in the community and I fear that the proposed method would not become a standard because of the increased memory cost (for instance compare max seq len possible for B=1 for TabPFN v1 arch vs yours -using efficient attention kernels of course- for inference).
> >
> > I am positively inclined towards the paper, but I am not convinced this method will end up used because of the added memory cost, so because I am afraid impact will be limited, I elect to currently keep my current score.

---

> > > ### Author Response · Authors · 2025-08-07
> > >
> > > Thank you for this raising this important question!
> > >
> > > Long-context ICL models is indeed an important research direction that, we believe, is compatible with the design of target equivariance models.
> > > The limited context is mainly due to the use of a standard attention on data/row mechanism. Some recent works suggest ways to overcome such limitation. For instance, TabICL [1] introduces a column-then-row attention mechanism to build fixed-dimensional embeddings of rows, increasing the context size up to 10K. The model can adapt to arbitrary input dimensions but requires a fixed target dimension.
> > > A possible interesting direction would be to combine such mechanism with our target equivariant decoders. Recent work on linear attention may also remove this limitation as shown in TabFlex [2] for instance which can scales to millions of samples with efficient attention. This, along with the fact that target equivariance reduces the need for ensembling could further alleviate the challenges of additional memory/compute.
> > >
> > > Thank you again for the interesting question, we will make sure to include these points in the discussion.
> > >
> > >
> > > **References**
> > >
> > > [1]: Qu, J., HolzmÃžller, D., Varoquaux, G., & Morvan, M. L. (2025). Tabicl: A tabular foundation model for in-context learning on large data. ICML 2025
> > >
> > > [2]: Zeng, Y., Dinh, T., Kang, W., & Mueller, A. C. (2025). Tabflex: Scaling tabular learning to millions with linear attention. ICML 2025

---

### Official Review · Reviewer_RkbL · 2025-07-02

**Clarity:** 4
**Significance:** 4
**Originality:** 4
**Rating:** 5
**Confidence:** 2

**Summary:**

This paper introduces an equivariant model for tabular data to address the lack of target equivariance in existing methods. It does this by adding equivariance to the components of a TabPFN model, especially by adding a bi-attention mechanism in the backbone of the architecture. It demonstrates the superiority of the model over existing baselines in the case of unseen class counts at test time.

**Questions:**

- What is the difference in flops between TabPFN and EquiTabPFN?
- What is the benefit of the individual equivariance adding components in the encoder, backbone, and decoder? An ablation would be interesting to see here. It is possible that one component is more important than others and can help reduce the complexity of the method.

**Ethical Concerns:**

["NO or VERY MINOR ethics concerns only"]

**Final Justification:**

Following the authors' response, I will keep my original score of 5. The paper is interesting with sufficient experiments, but the computational overhead is large enough to not justify a higher score.

**Limitations:**

Yes

**Quality:**

4

**Strengths And Weaknesses:**

Strengths:
- The paper addresses an important challenge with an original methodology.
- It presents a theoretical argument for their approach, and shows that target equivariant models are optimal. It shows that models not inherently designed to be equivariant like so are either less robust or use more computational power to achieve equivariance.
- It empirically shows that the proposed model is more robust to the number of target classes, performing better than current models in the case that test classes differ from pre-training.



Weaknesses:

The paper already addresses the limitation that the attention mechanism adds quadratic complexity. In the interest of fairness, they compare to baselines with similar number of model parameters. But in keeping with LLM literature, it would be a fairer comparison to compare compute flops to baselines, especially given that their method adds more complexity not necessarily reflected in the amount of parameters.

---

> ### Author Rebuttal · Authors · 2025-07-30
>
> Thank you for your positive feedback. To address your points, we performed the FLOPS and ablations comparison and summarize the findings below.
>
> **Flops comparaison.** While EquiTabPFN and TabPFN have similar parameter counts, EquiTabPFN uses more FLOPS. On an A100 GPU with 2,000 samples (100 features, 10 classes), EquiTabPFN required 566 GFLOPS vs. 76 GFLOPS for TabPFN (~7.45× more). However, with more classes (15), the gap narrows due to the ensembling needed for TabPFN to handle more than 10 classes (EquiTabPFN: 820 GFLOPS; TabPFN: 456 GFLOPS), consistent with runtime trends.
>
> Still, FLOPS or runtimes need to be traded for performance as in Figure 5 to which we added TabPFNv1 as requested by reviewer gnTN. There, EquiTabPFN offers the best trade-off between time and performance in most cases as summarized in the table below, in particular since TabPFN requires more ensemble to reach good performance. Finally, we note those numbers remain small for a modern GPU given that a single H100 can easily reach 400 TFLOPS on an LLM training workflow for instance.
>
> *Table of time/performance tradeoff:*
> | Pareto points   | 1        | 2        | 3        | 4        | 5        | 6        | 7        |
> |---------|----------|----------|----------|----------|----------|----------|----------|
> | Old Pareto front   | LinearModel     | EquiTabPFN(1)       |___________| EquiTabPFN(4)      | EquiTabPFN(12)      | EquiTabPFN(18)        | ___________ |
> | New Pareto front | TabPFNv1(6)         | EquiTabPFN(1)       | TabPFNv1(18)       | EquiTabPFN(4)       | EquiTabPFN(12)        | EquiTabPFN(18)    | EquiTabPFN*(18)   |
> | Time| 0.269 | 0.439 | 0.789 | 1.978 | 3.230 | 4.907 | 9.979 |
> | Performance| 51.06 | 60.14 | 61.06 | 61.73 | 61.74 | 63.68 | 66.50 |
>
> **Ablations.** Thank you for the suggestion, we will include the following ablation results on different architectures in the revised version. Specifically, we report relative error reduction over TabPFN in two settings: (1) using the TabPFN backbone (without bi-attention) with our equivariant decoder, and (2) using a bi-attention backbone with a standard MLP decoder (as in TabPFNv1). As shown in the table below, the combination of both—biattention and the equivariant decoder, eg the EquiTabPFN model we propose —yields the largest performance gain. The encoder, a simple linear embedding, is considered part of the backbone: fully connected for TabPFN-style models and 1x1 convolution for biattention-based ones.
>
> | Method                                | % Error reduction over TabPFN  |
> |----------------------------------------|-------------|
> | EquiTabPFN (Bi-attention backbone + Equivariant decoder)                          | +1.50%      |
> | TabPFN backbone + Equivariant decoder   | +0.94%      |
> | Bi-attention backbone + MLP decoder                       | -0.12%      |

---

> > ### Comment · Reviewer_RkbL · 2025-08-04
> > **Response to rebuttal by authors**
> >
> > Thank you for the additional information. I believe that the flops comparison is needed in the paper, given that there is a very high overhead (~order of magnitude) for EquiTabPFN. Even for >15 classes, 2x is a significant overhead. Please add this experiment to the revision.

---

> > > ### Author Response · Authors · 2025-08-04
> > > **Answer**
> > >
> > > Thank you for your feedback, we will certainly add this in the camera ready.

---

### Decision · Program_Chairs · 2025-09-17

**Decision:**

Accept (poster)

**Comment:**

The submitted paper introduces EquiTabPFN, a modification to the TabPFN model, addressing the lack of target equivariance in existing methods classification. The authors propose a bi-attention mechanism  to achieve target equivariance. Empirical results show that EquiTabPFN outperforms existing baselines, particularly in scenarios where the number of test classes differs from training. The paper also highlights the trade-offs between computational cost and performance and relate EquiTabPFN and TabPFN in this regard.

**Strengths of the paper**
1. Significance and originality:
   - The paper addresses a limitation in TabPFN by introducing target equivariance, which is both theoretically justified and practically important.

2. Theoretical contributions:
   - The authors provide a theoretical analysis, building on the concept of an "equivariance gap" and show that models lacking equivariance are suboptimal.

3. Empirical validation:
   - Experiments on real-world and synthetic datasets demonstrate the superiority of EquiTabPFN in terms of robustness and performance.
   - The paper includes ablation studies to disentangle the contributions of different architectural components.

4. Clarity:
   - The paper is well-written, with clear explanations of the methodology, theoretical insights, and experimental results.

5. Rebuttal period improvements:
   - The authors addressed reviewer concerns by adding FLOPS comparisons, ablation studies, and detailed discussions on computational trade-offs.

**Weaknesses of the paper**
1. Computational overhead:
   - EquiTabPFN incurs a significant computational cost compared to TabPFN, particularly for datasets with fewer than 10 classes. For example, EquiTabPFN requires ~7.45× more FLOPS than TabPFN in some scenarios.
   - While the authors argue that the trade-off is justified by performance gains, this overhead may limit the method's scalability to larger datasets.

2. Limited generalization to non-equivariant data:
   - Some reviewers raised concerns about the applicability of target equivariance in cases where class order carries semantic meaning (e.g., ordinal classes).

3. Baselines:
   - The paper does not include a comparison with ModernNCA, a relevant baseline that employs a similar prediction layer. The authors did not include this baseline because it requires training on all datasets.

4. Memory constraints:
   - The increased memory requirements due to the use of 4D tensors may limit the model's applicability in certain settings.

**Discussion**
In the rebuttal period, the authors addressed most reviewer concerns:
- FLOPS and time/performance trade-offs: The authors provided detailed comparisons, showing that EquiTabPFN achieves better performance at higher computational costs. They also clarified that the method becomes similar efficient for TabPFN for datasets with more than 10 classes due to reduced ensembling requirements.
- Ablation studies: The authors demonstrated that both the bi-attention backbone and the equivariant decoder contribute to performance gains, with their combination yielding the best results. They will add the shown results and further to the revised version of the paper.
- Equivariance vs. non-equivariance: The authors discussed the potential to reintroduce non-equivariance through positional encodings.
- Scalability: The authors acknowledged the memory limitations and suggested that recent advances in efficient attention mechanisms could mitigate these issues.

**Recommendation**
The paper is technically solid and has relevant theoretical and empirical contributions. While the computational overhead is a concern, the authors provide convincing justifications for the trade-offs. The work is relevant to the field, particularly in scenarios requiring robustness to varying class counts. The reviewers unanimously recommended acceptance, and I am following their recommendation.